# Botulinum Toxin Type A for the Treatment of Skin Ulcers: A Review Article

**DOI:** 10.3390/toxins14060406

**Published:** 2022-06-14

**Authors:** Waranaree Winayanuwattikun, Vasanop Vachiramon

**Affiliations:** Division of Dermatology, Faculty of Medicine Ramathibodi Hospital, Mahidol University, Bangkok 10400, Thailand; wara.mim@gmail.com

**Keywords:** BoNT-A, ischemia, neuromodulators, neurotoxin, Raynaud’s phenomenon, wound, wound healing process

## Abstract

The normal biological wound healing process consists of three precisely and highly programmed phases that require optimal conditions including internal and external factors. Any negative factors that disrupt the sequence or time frame of the healing mechanism can result in a non-healing wound or chronic ulcers. Botulinum neurotoxin A (BoNT-A) which is generally known as anti-contraction of muscles has been reported as a successful treatment in various types of chronic ulcers. The aim of this study is to review the outcome of treatment with BoNT-A for chronic skin ulcers. The results demonstrated some positive effects of BoNT-A on chronic ulcers. Ischemic ulcers secondary to Raynaud’s phenomenon seem to be the most promising type of ulcers that have benefited from BoNT-A. The rationale behind using BoNT-A to fasten the wound healing process is also discussed. Further clinical trial studies should be conducted to affirm the efficacy of wound healing using BoNT-A administration.

## 1. Introduction

Normal physiologic wound healing happens in three highly integrated and overlapping phases: inflammation, proliferation, and remodeling [1,2,3]. Good blood circulation, proper immune function, and adequate nutrition are required for optimum healing [1,4]. When any phase is disrupted, healing might be impeded and turned into chronic ulcers which are generally known as loss of normal skin integrity for more than 6 weeks [5,6,7]. Many factors attribute to retard wound healing. For instance, vascular insufficiency, neurologic abnormalities, nutritional deficiencies, advanced age, chronic diseases, and local wound infection can all disrupt the healing process [1,8,9,10]. Not only removing primary underlying causes but also following principles of wound care such as appropriate tissue debridement and optimal moist environment for normalization of the wound healing process is essential [1,11,12]. However, unsatisfied outcomes still exist [6,13,14]. 

Botulinum toxin is a neurotoxin produced by *Clostridium botulinum* [15]. It has long been known for its action in preventing the release of the neurotransmitter acetylcholine from axons at the neuromuscular junction and temporally inhibiting muscle contraction. So far, there are seven serotypes (A, B, C1, D, E, F, and G) [16]. Serotypes A and B are currently in clinical usage, but botulinum neurotoxin A (BoNT-A) is the most commonly used [15,16]. Concerning temporary muscle paralysis capability, wide clinical treatment has been adopted [16,17,18]. Apart from inhibiting acetylcholine release from a presynaptic neuromuscular junction, studies found that BoNT-A can reduce skin inflammation by inhibiting mast cell degranulation and blocking cholinergic stimuli to apocrine and eccrine glands [19]. 

Recent studies found that BoNT-A has the efficacy to promote wound healing by attenuating the release of norepinephrine and many neurotransmitters which inhibit vasoconstriction and increase blood flow [20,21]. The inhibitory effect of several pain-related neurotransmitters has also been mentioned [22,23,24]. 

Herein, this review aims to focus on the positive effects of BoNT-A on the wound healing process and the outcome of BoNT-A for the treatment of skin ulcers from various etiologies.

## 2. Methods

A literature search was performed in PubMed database for articles published in English before 31 December 2021. The articles that employed BoNT-A for ulcers regardless of the etiology were included. Search keywords were botulinum neurotoxin type A, botulinum toxin, botulinum toxin therapy, neurotoxin, neuromodulator, ulcer, chronic skin ulcer, wound, and wound healing. Combination search terms were “botulinum toxin” with “ulcer”, “chronic skin ulcer”, “wound”, or “wound healing”. The selection criteria were studies that used BoNT-A injection for chronic wounds or ulcers regardless conditions. We Excluded articles written in other languages than English or using botulinum neurotoxin B injection.

According to our PubMed search, 20 articles were obtained and reviewed. Twelve articles were ulcers secondary to Raynaud’s phenomenon (RP), whereas eight articles were other types of chronic ulcers namely pressure ulcers, traumatic ulcers, and so forth. In terms of the type of articles, nine studies were case report, five studies were retrospective case series, one study was prospective case series, and two studies were case series. Others were two letters to editors and one correspondence. Due to most studies being case reports and small case series, a systematic review could not be established. Therefore, we reviewed and discussed based on available data. 

## 3. Mechanism of BoNT-A on Wound Healing

Normal physiologic wound healing is a dynamic process consisting of different continuous, overlapping, and precisely programmed phases [1,3,11]. Any disruption in the process leads to abnormal wound healing or chronic unhealed ulcers [12]. Our literature review found BoNT-A can enhance wound healing in various types of ulcers. Several studies that investigated mechanisms of BoNT-A on wound healing found that BoNT-A decreased inflammatory cell infiltration during the inflammatory phase [25,26,27]. Inhibition of mast cell degranulation and reduction in number were reported [19,28]. Reduction in vascular permeability and exudation of neutrophils and macrophages were also observed. Consequently, pus deposition on ulcers was reduced [29]. Furthermore, during the proliferative phase, it could enhance blood flow and promote vascular sprouting by increasing expression in CD31+, ∝SAM (+) pericytes, as well as fibroblasts [30,31,32,33]. It could be due to a reduction of reactive oxygen species (ROS) release and endothelial nitric oxide synthase (eNOS) in the hypoxic area [30,34]. Altogether, granulation tissue was created and filled the wound. It also can inhibit releasing norepinephrine resulting in vasodilatation and increasing oxygenation to the wound [20]. In the period of the remodeling phase, lower expression of TGF-β1 by BoNT-A was found which reduces the risk of fibrosis and scar formation [29]. Figure 1 shows a mechanism of BoNT-A in wound healing for comprehensive understanding.

For miscellaneous types of chronic ulcers, BoNT-A has been proposed to hasten wound healing by inhibiting sweat-induced maceration of the fragile epidermis to optimize the wound environment [35]. 

## 4. BoNT-A for Various Types of Skin Ulcers

### 4.1. Ischemic Ulcers Secondary to Raynaud’s Phenomenon (RP)

Although numerous publications demonstrated a positive effect of BoNT-A on RP [36,37,38,39,40], this review article will mainly focus on patients who had RP-associated ulcers. Based on the literature search, 12 articles (case reports, retrospective case series, and prospective case series) with a total of 104 patients who had RP-associated ulcers were identified (Table 1) [41,42,43,44,45,46,47,48,49,50,51,52]. Patients have been suffering from RP symptoms (pain, loss of function, disfigurement, and so forth) and chronic ischemic nonhealing ulcers. Moreover, some underwent sympathectomy but the clinical of ulcers did not improve [41]. 

There were seven studies that reported the type of BoNT-A used [42,43,46,48,49,50,52]. Onabotulinum toxin (Botox^®^, Allergan Pharmaceuticals Ltd., Westport, Ireland) was identified in five studies [42,43,46,49,50] and Medytoxin^®^ (Medytox, Seoul, Korea) was used in two studies [48,52]. Regarding onabotulinum toxin, a minimum dose was 10 units, and a maximum dose was 77 units per one affected area [43,46]. Two reports from Korea used Medytoxin^®^ with the same dose of 10 units [48,52]. Other articles did not mention the type of BoNT-A or toxin brand with the dose of 32–100 units per 1 affected area [41,44,45,47,51]. In terms of reconstitution, most common reconstitution adopted was toxin 20 units to 0.9% NSS 1 mL [45,46,49,53]. Two reports from Korea used BoNT-A 100 units to 0.9% NSS 1 mL [48,52]. The toxin concentration of other studies was BoNT-A 50 units to 0.9% NSS 1 mL [41,43] and BoNT-A 5 units to 0.9% NSS 1 mL [50], respectively.

There was no standard guideline regarding the site of injection. Nevertheless, three areas of injection were proposed to target neurovascular bundles on palms: (1) Base of digits at web spaces (i.e., bifurcation of the superficial digital arteries) [41,42,43,49,51]; (2) Palmar aspect of the hand, just proximal to the A1 pulley, targeting the neurovascular bundles [41,46,50]; and (3) Both sides of proximal hand (i.e., radial and ulnar arteries) [42,43]. Most studies adopted either option 1 or 2 or a combination [41,46,49,50,51]. Two retrospective studies injected all three areas [42,43]. Injection patterns are shown in Figure 2. There were two reports with foot ulcers injected at interdigital web space oriented toward the neurovascular bundle [44,45]. 

Parameters tracked outcomes consisted of objective and subjective measurements. Among objective measurements, arterial blood flow was evaluated by using ultrasonography or doppler perfusion imaging, while skin surface temperature was assessed by a thermometer or temperature recovery after ice-bath immersion [46]. In terms of subjective assessment, a visual analog scale (VAS) for pain, general RP symptoms using Raynaud’s score [54], digital color changes, and patient global assessment were used. However, for healing ulcers, no wound assessment scale was used. Investigators merely monitored ulcers as complete, partial healing, and no response.

Of a total of 104 cases with RP-associated ulcers, 81% (84 of104 cases) had healed completely. Immediate improvement of blood flow, pain, temperature, and color after injection was noted [41,45]. Raynaud’s score and VAS for pain decreased at 2 weeks and persisted until 16 weeks after injection [46]. 

One case report presented a 55-year-old woman who was diagnosed with limited cutaneous systemic sclerosis with refractory multiple digital ulcers. She failed many medical treatment regimens and underwent amputation once. During the period of critical ischemic digits which further amputation was nearly executed, BoNT-A was challenged, and digits were rescued with complete wound healing eventually [47]. Another case report showed a 48-year-old male who was diagnosed with anti-MDA5-Ab-positive dermatomyositis with refractory digital ulcers had an improvement in ulcers after BoNT-A injection [48]. 

According to a 3-year follow-up study by Medina et al., 8 of 14 patients (57.1%) demonstrated a very good response at 1 month after treatment. A mild to moderate response was observed in three patients (21.4%). Of the seven patients with basal ulcers, five were completely healed at 3 months after treatment. At the end of treatment, 64.3% of patients showed an overall satisfaction level of >8 [49]. 

However, some patients with RP showed no response to BoNT-A injection which resulted in amputation. Therefore, the discussion with patients before employing this technique is mandatory [43,49,50]. 

### 4.2. BoNT-A for Pressure Ulcers

Administration of BoNT-A on pressure ulcers that were associated with muscle spasticity has been reported [55,56,57,58]. The rationale for the use of BoNT-A on pressure ulcers is to relieve muscle spasticity [59,60]. As a result, pressure ulcers are adequately accessed and omitted from repetitive trauma. 

Regarding the point of injection, BoNT-A can be injected directly into abnormally contracted muscle or under electromyographic guidance [55,56,57]. The number of injection points was considered based on the size of muscles. According to a report by Insito et al., abobotulinum toxin with doses of 200 and 120 speywood unit (SU) was injected into orbicularis oris and masseter in a vegetative state patient with oromandibular dyskinesia [57]. Regarding the larger size of muscles such as Gluteal muscle, a high dosage of 660 SU was used [56]. Gupta and Wilson reported a dosage of onabotulinum toxin ranging from 100 to 150 units per muscle [55]. Abnormal contraction or spasticity were improved as early as 1 week after treatment [5]. All ulcers were reported as complete healing with the most delayed time period of 6-month follow-up [56]. The number of treatment sessions varied from one to two sessions. A repeated treatment session might be considered in patients with partially healed ulcers to maintain the weakness of muscles. Data regarding the use of BoNT-A in pressure ulcers are summarized in Table 2.

### 4.3. BoNT-A for Traumatic Ulcers

Posttraumatic ulcers associated with vascular compromise (e.g., crush, direct drug injection, proximal arterial injury from catheterization, etc.) were reported successfully treated by BoNT-A [61,62]. According to a retrospective cohort study by Laarakker et al., patients with traumatic ischemic ulcers were categorized into two groups (BoNT-A treated group vs. non-BoNT-A group). In the BoNT-A treated group, 80 to 100 units of onabotulinum toxin were injected into the palm of each patient. The location of injection was at the level of the distal palmar crease and close to the radial and ulnar arteries. All digits (100%) were rescued for the BoNT-A injection group, while 83% had amputation of necrotic digits in patients without BoNT-A injection [61]. In addition, pain scores were lower in BoNT-A-treated fingers when compared to no BoNT-A injection. The postulated mechanism was the improvement of blood flow by BoNT-A. Another traumatic ulcer on the hand of a 4-year-old child was rescued without amputation by administration of 10 units BoNT-A into the proximal palm, the radial and ulnar artery locations of the distal forearm [62]. Data regarding the use of BoNT-A for traumatic ulcers are summarized in Table 2.

### 4.4. BoNT-A for Other Types of Chronic Ulcers

Neuropathic foot ulcer has been reported to be successfully treated by two sessions of BoNT-A injection [63]. The amount of 70 units of onabotulinum toxin was infiltrated around the ulcer with a repeated injection at 3 months with a similar dosage. The proposed mechanism was a reduction of sweat-induced maceration and optimization of the wound-healing environment.

Zhong et al. reported various types of chronic ulcers that have been successfully treated by BoNT-A administration in four patients [53]. One interesting case was a chronic infective skin ulcer on the left temporal region due to acne squeezing. After systemic antibiotics and debridement, there was a slight improvement in the ulcer. Multiple points of BoNT-A injection around the ulcer were done with 32 units in total. The wound was completely healed 20 days after injection. Data regarding the use of BoNT-A for other types of chronic ulcers is summarized in Table 2.

## 5. Practical Guidelines for Treatment

To date, a standard guideline for BoNT-A injection for chronic ulcers has not been established yet due to a lack of strong evidence that supports the efficacy of BoNT-A for skin ulcers. Based on the available data, BoNT-A might be offered to patients with chronic skin ulcers due to vascular compromise (i.e., RP-associated ulcers, pressure ulcers with vascular compromised), traumatic ulcers, etc. (Figure 3). BoNT-A for RP-associated ulcers seems to be the most promising efficacy and established treatment method. Nevertheless, its harmlessness and ubiquity make it worth trying for those chronic ulcers that failed standard therapy. Adverse effects were mild and temporally, an intrinsic hand muscle weakness has been reported which resolved completely within 5 months [41,43,49,50]. In terms of point of injection, our recommendation is to consider the type of ulcer including (1) ischemic ulcers: inject toward neurovascular bundles for vasodilation [41,42,43,46,49,50,51]; (2) pressure ulcers: inject into contracted and spastic muscles for muscle relaxation; and (3) other types: inject around the ulcer to optimize wound environment [53,63]. Dosage and reconstitution should be considered individually depending on wound size or volume of muscle. 

## 6. Conclusions

In summary, BoNT-A injection, a minimally invasive procedure that has a low rate of side effects can be adjunctive therapy for enhancing wound healing in various types of chronic ulcers that have been treated for underlying causes and had wound care properly as well as in ischemic ulcers associated RP in which failed conventional therapy. However, there is no randomized controlled trial study (RCT) with a large number of patients to affirm those efficacies. The amount of BoNT-A injection and the exact point of injection is still uncertain. Future randomized controlled studies should be conducted to evaluate the efficacy and safety of BoNT-A for various types of ulcers with different anatomical regions.

## Figures and Tables

**Figure 1 toxins-14-00406-f001:**
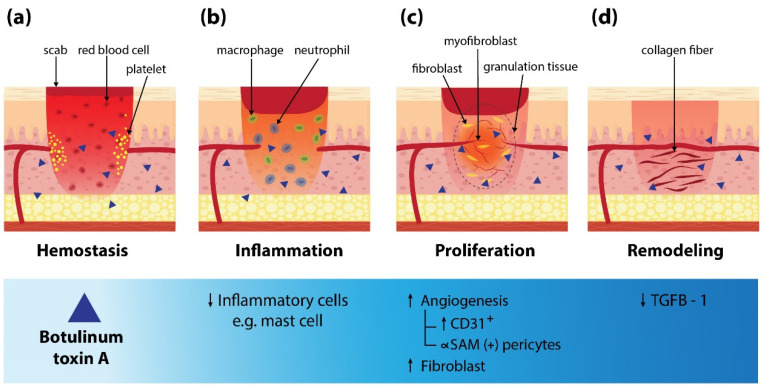
The proposed mechanisms of botulinum neurotoxin A in the wound healing process. TGFβ-1: Transforming Growth Factor β-1.

**Figure 2 toxins-14-00406-f002:**
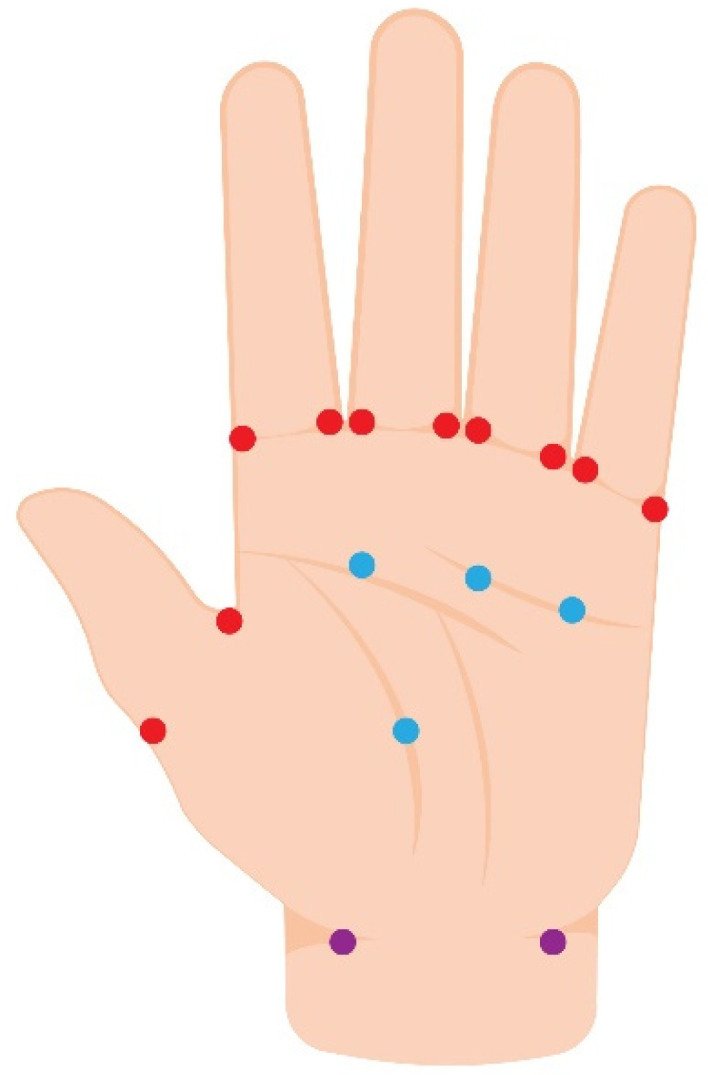
The BoNT-A injection patterns for Raynaud’s phenomenon-associated ulcer. Red dots represent the base of the digit injection pattern. Blue dots identify the palmar injection pattern. Purple dots identify the proximal hand injection pattern.

**Figure 3 toxins-14-00406-f003:**
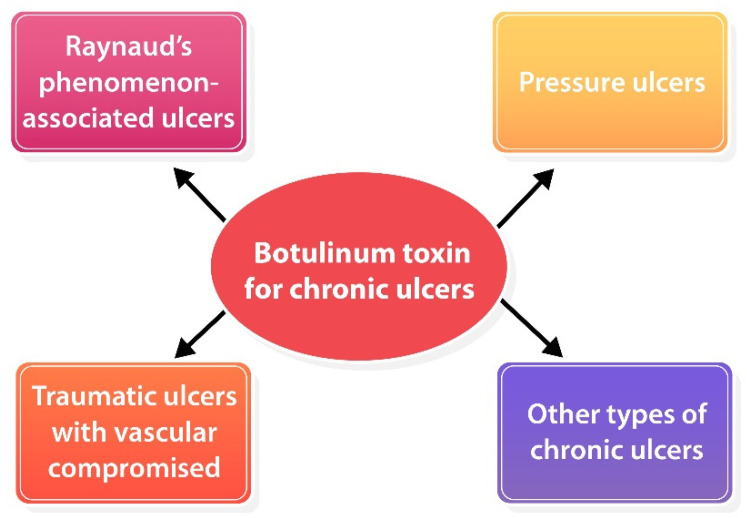
Potential role of BoNT-A for various types of chronic ulcers.

**Table 1 toxins-14-00406-t001:** A summary of articles on the treatment outcome of chronic ischemic ulcers secondary to Raynaud’s phenomenon.

Authors, Year	Study Type	N (Gender)	Age, Years (Mean or Range)	Type of BoNT-A	BoNT-A Dilution with 0.9% (NSS)	BoNT-A Dose/Location	Follow-Up Period	Results	Reinjection (Interval)	Comments
Quinatana Castanedo et al., 2020 [44]	Case report	1 (female)	15	NR	NR	32 units/Foot	4 weeks	- Complete resolution - Complete epithelization	Yes (every 12 months)	
Habib et al., 2020 [51]	Case series	3 (female)	23–50	NR	NR	32 units/Hand	1 week	- 75% (2/3) completely free of symptoms - No effect (1 case)	No	
Min et al., 2020 [48]	Case report	1 (male)	48	Medytoxin® (Medytox, Seoul, Korea)	10 units/0.1 mL	10 units/Hand	12 weeks	- Healed	Yes (weekly for 3 weeks)	
Souk and Kim, 2019 [52]	Case report	2 (female)	50 and 62	Medytoxin® (Medytox, Seoul, Korea)	10 units/0.1 mL	10 units/Hand	8 weeks	- Healed and almost healed	Yes 2/2 (4 and 5 weeks, 7 and 8 weeks)	
Garrido-Rios et al., 2018 [45]	Case report	1 (female)	30	NR	8–10 units/0.4 mL	80–100 units/Hand	2 months	- Reduction in necrotic area	No	
Medina et al., 2018 [49]	Retrospective case series	15 (female 14/male 1)	35–71	Botox® (Allergan Pharmaceuticals Ltd., Westport, Ireland)	100 units/5 mL	Average 45 units/Hand	3 years	- Significantly decreased in 1 month - Significantly decreased in 1 month - 71% (5/7) patients healed at 3 months	Yes 6/15 (annually)	4/15 temporary decrease intrinsic muscle strength
Blaise et al., 2017 [47]	Case report	1 (female)	55	NR	NR	100 units/Hand	4 months	- Completely healed - Increased skin blood flow	No	
Motegi et al., 2016 [46]	Prospective, case series	10 (NR)	62.5 (±3.5)	Botox® (Allergan Pharmaceuticals Ltd., Westport, Ireland)	20 units/0.1 mL	10 units/Hand	16 weeks	- Significant reduction at 2 weeks and throughout the study - Significant reduction at 2 weeks and throughout the study - Significantly enhanced at 4 weeks - 100% (5/5) healed within 12 weeks	No	
Zhang et al., 2015 [42]	Retrospective case series	10 (female 5/male 5)	48–91	Botox® (Allergan Pharmaceuticals Ltd., Westport, Ireland)	100 units/5 mL	60 units/Hand	6 months (average)	- 100% Improvement, significant improvement of the PSV - Significantly increased on palms and fingers - Significant decrease in pain, numbness, stiffness, swelling - 50% (1/2) healed at 3.5 weeks	No	
Smith et al., 2012 [41]	Case report	1 (female)	52	NR	5 units/0.1 mL	100 units/Hand	3 months	- Improvement blood flow - Decrease - 67% (6/9) healed - 22% (2/9) partially healed	No	Mild, nonlimiting thenar muscle weakness
Neumeister. 2010 [50]	Retrospective case series	33 (female 19/male 14)	18–72	Botox® (Allergan Pharmaceuticals Ltd., Westport, Ireland)	100 units/20 mL	50 units/Hand	6 years	- 85% (28/33) relieved - Improvement in perfusion - 100% healed within 2 months	Yes 7/33 (not reported)	- 3 patients had temporary intrinsic muscle weakness that lasted 2 months
Fregene et al., 2009 [43]	Retrospective case series	26 (female 14/male 12)	60.7 (±1.9)	Botox® (Allergan Pharmaceuticals Ltd., Westport, Ireland)	100 units/2 mL	Average 77 units/Hand	18 months (average)	- Significant mean 35% reduction - Significant color improvement in the female and smoker’s subgroup - Significant increasing - 48% (11/23) healed (average healing time 9.5 weeks)	No	- Some reported intrinsic muscle weakness and 1 dysesthesia digit which resolved completely by 5 months

Abbreviations: MRA; Magnetic resonance angiography, NR; Not reported, NSS; Normal saline, PSV; Peak systolic velocity, SD; Standard deviation, VAS; Visual analog scale.

**Table 2 toxins-14-00406-t002:** A summary of articles on the treatment outcome of other types of chronic ulcer.

Authors, Year	Study Type	N (Gender)	Age, Years (Mean or Range)	Type of BoNT-A	BoNT-A Dilution with 0.9% NSS	BoNT-A Dose/Location	Follow-Up Period	Results	Reinjection (Interval)	Comments
Gupta and Wilson, 2020 [55]	Case report	1 (female)	59	NR	NR	150 units for pectoralis major, 150 for elbow flexors, 100 for flexor digitorum superficialis	5 months	Completely healed ulcer	Yes (5 months)	Pressure ulcer
Insito and Basciani, 2009 [56]	Case report	1 (male)	27	Dysport^®^, Ipsen Limited, Slough, UK	NR	660 Speywood units (left Gluteus maximus)	6 months	Weaken muscle contraction Healed ulcer	Yes (3 months)	Pressure ulcer
Insito et al., 2008 [57]	Case report	1 (male)	73	Dysport^®^, Ipsen Limited, Slough, UK	NR	200 Speywood units for Orbicularis oris, 120 for Masseter	3 months	Improved dyskinetic disorder Completely healed ulcer	Yes (2 months)	Pressure ulcer
Sillitoe et al., 2007 [58]	Letter to editors	1 (male)	58	NR	NR	NR (adductor muscle bellies lower limbs)	16 weeks	Marked reduction in spasticity Ulcers showed signs of healing Ulcers show significant improvement Ulcers fully healed	No	Pressure ulcer
Laarakker and Borah, 2020 [61]	Retrospective cohort, case series	5 (NR)	31–71	NR	NR	80–100 units (palm and wrist)	NR	All Digits were preserved	No	Traumatic ulcer
Upton et al., 2009 [62]	Letter to editors	1 (NR)	4	NR	NR	10 units (palm)	NR	The digits were rescued	No	Traumatic ulcer
Zhong et al., 2019 [53]	Case series	4 (female 1/male 3)	16–78	NR	NR	32–48 units (face, leg, foot)	50 days	Ulcers healed	No	Chronic skin ulcer
Alsharqi et al., 2011 [63]	Correspondence	1 (male)	51	Botox^®^ (Allergan Pharmaceuticals Ltd., Westport, Ireland)	NR	70 units (right foot)	3 months	Completely healed ulcer	Yes (3 months)	Neuropathic ulcer

Abbreviation: NR; Not reported, NSS; Normal saline.

## Data Availability

The raw data supporting the conclusions of this article will be made available by the authors, without undue reservation, to any qualified researcher.

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
