# Peer review of "Botulinum Toxin Type A for the Treatment of Skin Ulcers: A Review Article"

_toxins, 2022, doi:10.3390/toxins14060406_

Round 1

Reviewer 1 Report

In this manuscript, the authors reviewed the outcome of a treatment with botulinum neurotoxin A (BoNT-A) for chronic skin ulcers.

This manuscript is interesting; unfortunately, the manuscript needs substantial improvements and corrections before publishing may be possible.

General points:

Please say in the whole manuscript: botulinum neurotoxin A (BoNT-A).

Please add a list of abbreviations before References section to your manuscript.

English language correction by the native speaker is required.

Please add all citations (references number) in the whole text according to “Toxins”. 

Special points: 

For better readability, please add more Figures to your manuscript.

Keywords: please add also to keywords: Wound healing process; BoNT-A.

Important, this manuscript should be substantially improved, i. e., by substantial references in the field.

 Introduction

 Lines 23-24: please add multiple references at the end of this sentence.

 Lines 24-25: please add multiple references at the end of this sentence.

 Lines 25-28: please add multiple references at the end of this sentence.

 Lines 30-32: please add multiple references at the end of this sentence.

 Lines 34-42: please add multiple references at the end of each of these sentences.

 2. Mechanism of BTX-A on wound healing

 Lines 52-58: please add multiple references at the end of each of these sentences.

 Lines 61-64: please add multiple references at the end of this sentence.

4. BTX-A for various types of skin ulcers 80

4.1. Ischemic ulcers secondary to Raynaud’s phenomenon (RP)

 Lines 92-94: please add multiple references at the end of each of these sentences.

 Lines 102-106: please add multiple references after “1”, after “2” und after “3”.

 Lines 106-107: please add multiple references at the end of this sentence.

 Lines 110-115: please add multiple references at the end of each of these sentences.

 Lines 118-119: please add multiple references at the end of this sentence.

4.2. BTX for pressure ulcers

 Lines 139-140: please add multiple references at the end of this sentence.

 Lines 142-153: please add to your review multiple Figures demonstrating the injection points of the BoNT-A into the different muscles described by you in all these sentences.

 5. Practical guideline for treatment

 Lines 191-196: please add the references after „1“-„3“.

 Table 1: please add the exact number of the female and male patients in each publication and add this information into the column “3”.

 Table 2: please add the exact number of the female and male patients in each publication and add this information into the column “3”.

References

 Please prepare your References list according to the instructions of “Toxins”.

Reviewer 2 Report

I have no objections to this mini-review.  The authors have touched on an important and difficult topic, since there are no highly effective protocols in this area that guarantee a positive outcome, any attention to it is very important. It is encouraging that the works cited as a whole offer hope for the development of a proper treatment protocol. And as the authors mentioned "Future randomized controlled studies should be conducted to evaluate the efficacy and safety of BTX-A for various types of ulcer with different anatomical regions". 

Author Response

Thank you for your valuable comments. We strongly believe this article would be beneficial to the readers.

Reviewer 3 Report

The manuscript entitled “Botulinum toxin type A for the treatment of skin ulcers: a review article” broaches a remarkable task for which botulinum toxin treatment can provide positive outcomes. The manuscript is well written, and I would like to recommend its publication after a revision; however, I would like to suggest some possible improvements. 

Usually, the acronym used for botulinum neurotoxin is BoNT. Please replace BTX with BoNT.

The paragraph “2. Mechanism of BTX-A on wound healing” should be moved after the section “3. Methods”. In methods, the authors should describe in detail also the combination (if used) of the keywords reported in the text to allow the reader to replicate the bibliographic search and the criteria for inclusion/exclusion.

The authors should consider the opportunity to include a new paragraph describing the results of the bibliographic search in the manuscript. This paragraph should mention the number of manuscripts obtained, summarizing the main findings obtained and detailed in the other paragraph of the manuscript. This section should discuss their choice for the type of review carried out, underlining why the best option was the review and not a systematic or scoping review.

The authors should improve the paragraph “5. Practical guideline for treatment”, including details about the cases in which this treatment is recommended and inadvisable.

Round 2

Reviewer 1 Report

Thank you for your corrections.

You followed most of me suggestions.